# Classification of Lentigo Maligna at Patient-Level by Means of Reflectance Confocal Microscopy Data

**Romain Cendre** [1,*], **Alamin Mansouri** [1], **Jean-Luc Perrot** [2] **and Elisa Cinotti** [3] **and Franck Marzani** [1]

[1]   ImViA EA 7535, University of Bourgogne Franche-Comté, 21078 Dijon, France; alamin.mansouri@u-bourgogne.fr (A.M.); franck.marzani@u-bourgogne.fr (F.M.)
[2]   Dermatology Department, University Hospital of Saint-Etienne, 42055 Saint-Etienne CEDEX 2, France; j.luc.perrot@chu-st-etienne.fr
[3]   Department of Dermatology, Division of Medical, Surgical and Neuro-Sciences, University of Siena, Le Scotte Hospital, Viale Bracci 16, 53100 Siena, Italy; elisacinotti@gmail.com
[*]   Correspondence: romain.cendre@gmail.com



**Featured Application: This paper focuses on improvement in patient care and it also helps practitioners optimize their dermatology services by means of computer-assisted diagnostic software using data from reflectance confocal microscopy devices.**

**Abstract:** Reflectance confocal microscopy is an appropriate tool for the diagnosis of lentigo maligna. Compared with dermoscopy, this device can provide abundant information as a mosaic and/or a stack of images. In this particular context, the number of images per patient varied between 2 and 833 images and the objective, ultimately, is to be able to discern between benign and malignant classes. First, this paper evaluated classification at the image level, with the help of handcrafted methods derived from the literature and transfer learning methods. The transfer learning feature extraction methods outperformed the handcrafted feature extraction methods from literature, with a $F_1$ score value of 0.82. Secondly, this work proposed patient-level supervised methods based on image decisions and a comparison of these with multi-instance learning methods. This study achieved comparable results to those of the dermatologists, with an AUC score of 0.87 for supervised patient diagnosis and an AUC score of 0.88 for multi-instance learning patient diagnosis. According to these results, computer-aided diagnosis methods presented in this paper could be easily used in a clinical context to save time or confirm a diagnosis and can be oriented to detect images of interest. Also, this methodology can be used to serve future works based on multimodality.

**Keywords:** computer-assisted diagnosis; classification; transfer learning; reflectance confocal microscopy; dermatology; lentigo

---

## 1. Introduction

As the incidence rate for skin cancers has steadily increased over the years, they are now the most prevalent form of human malignancy. These diseases affect people in their everyday life, as they have a pronounced social impact on the affected individuals as a result of a decrease in their quality of life as well as because they have the potential to become lethal. In addition, they have significant economic consequences, with an estimated cost of 8 billion dollars per year in the United States [1], and they can stretch the ability of dermatological centers to cater to the at times overwhelming demand for screening and treatment. However, most of these repercussions can be avoided by early detection and appropriate surgeries [1].

Currently, clinicopathological correlation is the gold standard to diagnose skin cancers. The histological examination is relatively time-consuming as it requires excision of the affected area, the embedding of the sample in paraffin, the generation of thin tissue sections that then need to be stained and examined by a histopathologist. Despite its accuracy, this technique remains time-consuming, invasive, and inconvenient for doctors and patients. Consequently, several non-invasive imaging techniques have been developed to help the clinical diagnosis of skin cancers, and some of these are now commonly used by dermatologists. For instance, clinical photography and dermoscopy are both examples of affordable and intuitive techniques that are presently widely used by dermatologists. Dermoscopy tends to replace clinical photography as it significantly improves the quality of the diagnoses made by experts, due largely to the acquisition of high-magnification images of the skin [2].

Research papers on dermatology nowadays tend to focus on the dermoscopy modality used to perform automatic classification of lesions. Most of them obtain acceptable results with melanocytic pathologies [3]. Older methods focus on finding the most pertinent combination of preprocessing steps and handcrafted features to be used in a machine learning scheme [4,5]. By contrast, most recent methods use deep learning approaches, and they have yielded impressive results in this discipline [6]. In this particular study, the authors used an Inception-V3 architecture pre-trained on the "ImageNet" database [7], and they fine-tuned this model on a dataset of 129,450 clinical images containing 2032 different skin lesions and distributed across 757 classes. They carried out the classification at different taxonomy levels, and at the first level of classification (Non-neoplastic versus benign versus malignant), they achieved an accuracy of $0.72 \pm 0.9$ compared to 0.66 on a subset of these data by specialists.

However, dermoscopy imaging devices only provide surface and chromatic information. To overcome this limitation, reflectance confocal microscopy (RCM) modality is another type of imaging technique used by dermatologists that provides high-resolution images of the skin on a micrometer scale. Furthermore, this modality can provide structural information at different depths of the skin by adjustment of the wavelength properties and the focal point [8]. The RCM device was first designed by Marvin Minsky [9]. The principle of this device is to emit and focus a low power laser on a specific point of the skin, then the light from this spot is reflected and collected through an objective and a pinhole that allows only the light from the in-focus plane to reach the detector (see Figure 1). In this situation, the illuminated point and the detector aperture have confocal (a contraction of conjugate focal) planes [10]. The main interest of this focal point/plane is to provide only tissue information from a specifically chosen depth. Different factors can affect the depth: illumination wavelength, illumination power, reflective and scattering properties of the skin.

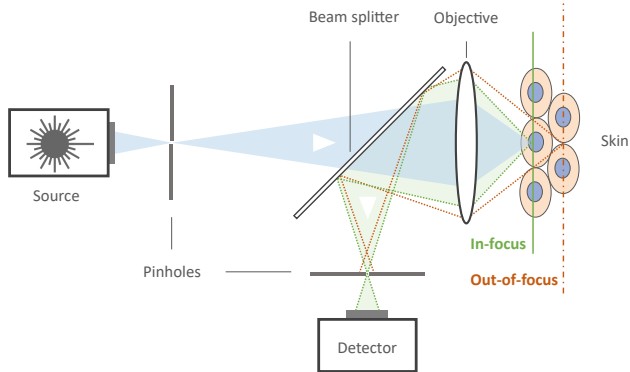

**Figure 1.** Principle of the RCM designed by Marvin Minsky [9]. The light source is transmitted through a pinhole and focused on the sample through an objective. Then the in-focus reflected light is collected by the detector with the help of the second pinhole.

Presently, this tool has greater diagnostic accuracy compared to dermoscopy, both for melanocytic and for non-melanocytic skin tumors [11–13]. Unlike the previous modalities, RCM remains expensive, although the number of users continues to increase [14] and recent developments have led to a degree of improvement in the portability of RCM devices [15].

By contrast, relatively fewer studies have been published on the RCM modality by use of computer vision techniques, despite their promising results in the clinical context. The use of artificial intelligence could be particularly useful for RCM images because their evaluation by dermatologists is time-consuming. Indeed, unlike with dermoscopy, many RCM images need to be acquired for each skin lesion. Many studies of computer vision focus on understanding these images by predicting the position in the skin layer [16,17] using stacks of 3D data provided by this modality. A number of other studies have described the structural components of the skin [18] and few of them have classified pathologies based on that specific modality. One of these studies [19] is quite similar to the problem explored in this work, and it suggests several image descriptors for undertaking a classification task. These authors introduced two methods using frequency representation: the first one based on Fourier transform and the second one based on wavelet representation by use of Daubechies 4. The idea behind spectral representation is to extract information at different frequency levels, thereby yielding information about smoothness or complex structures inside the images, and to be consistent with rotations and translations as the pathologies are non-oriented. The second category of descriptors that they employed was based on spatial features by use of gray level histogram (GLH) and gray level co-occurrence matrix (GLCM) statistical descriptors. The statistical descriptors computed from GLCM had been derived from previous work [20]. Finally, the authors proceeded with the classification of images by using different image parts at multiple sizes and they proceeded to classification and regression trees (CART) classification. In a two-class situation, this paper reached an accuracy of 0.96 for the detection of nevi and 0.97 for melanoma pathologies by applying a wavelet extraction and CART classification on a subpart of the $256 \times 256$ pixel images. Another paper is more relevant, as the authors followed the same purpose as this work by suggesting a way to classify solar lentigo pathologies by use of the previous wavelet decomposition, and by fitting the decomposition values to a generalized Gaussian distribution (GGD) to reduce the number of variables. They also suggested that only one variable and only one scale decomposition are relevant for solar lentigo detection. This method was applied to 45 subjects with healthy skin or solar lentigo, and it achieved a sensitivity of 0.81 and a specificity of 0.83 [21].

The scope of this work is to detect malignant tumors and particularly lentigo maligna/lentigo maligna melanoma (LM/LMM) (the most common type of facial melanoma) in RCM images and to help specialists reach a diagnosis based on these images at the patient-level. The previous feature extraction methods and extraction through different convolutional neural network (CNN) architectures were investigated first. As wavelet decomposition and reduction through GGD [21] were shown to be irrelevant in our data context [22], this work did not focus on any of these methods. A comparison of several classification models on full-size images was then carried out to estimate the relevance of these methods. Also, this study will employ the term "lentigo maligna" instead of the term "melanoma in situ type lentigo maligna" as it was used by the last RCM clinical studies [23–26] and for the sake of simplicity.

The following parts of this paper are organized as follows. Section 2 covers the data by proving details about their composition, the feature extraction methods implemented, and the process used to compute image-level and patient-level decisions. Section 3 then displays all of the results and provides an analysis of them, and finally, Section 4 provides a conclusion of this work and it offers a number of perspectives.

## 2. Materials and Methods

### 2.1. Data

The data for this paper were originally obtained in a previous clinical study [27] that performed a comparison between dermoscopy and RCM modalities in the diagnosis of benign lesions such as solar lentigo and malignant tumors by focusing essentially on LM/LMM. These images were acquired by three specialists, experts in regard to non-invasive skin imaging tools, by use of a hand-held VivaScope® 3000 camera that uses a laser with a wavelength of 830 nm and images to a depth of up to 250 µm. In addition, these data included lesions that can be highly misleading for dermatology specialists. From an ethical perspective, this study was conducted following the Declaration of Helsinki and the protocol was approved by the Ethics Committee of the University Hospital of Saint-Etienne (Institutional review board number 672016/CHUSTE).

Only images considered relevant for the diagnosis by two out of the three investigators were retained, at different depths of the skin: the epidermis, the dermal-epidermal junction (DEJ), and the dermis. However, most of the information was acquired at the DEJ. Each RCM image corresponding to a horizontal 920 µm × 920 µm section of the skin at a selected depth with a lateral resolution of 1 µm and axial resolution of 3 µm to 5 µm. For specifications, these images have a spatial resolution of 1000 × 1000 pixels with quantification on a single 8-bits channel.

The relative position between images of a single patient was unknown, and this could not provide any further knowledge in the next part of this work. Furthermore, the metadata available regarding the age of the patients was not used as the purpose of this work was to evaluate the relevance of image classification techniques, although it was initially provided to the experts during their assessments.

These data include 223 lesions from 201 patients, for a total of 7846 RCM images. Generally, one lesion is equal to one patient so we will discuss of a lesion as a patient in the next paragraphs. Each of these cases varies between 2 and 833 images (with a mean of 35 and a standard deviation of 64). For each patient, the data provided a diagnosis based on clinicopathological correlation that served as a reference basis and there was the following distribution of cases:

- 135 patients had "malignant" tumors: 115 LM/LMM and 20 basal cell carcinoma (BCC).
- 88 patients had "benign" tumors: mainly represented by solar lentigines.

No collision tumors were included in this series. Also, the study [27] evaluated 21 experts, and they achieved a mean sensitivity of 0.80 (range 0.66–0.90, standard deviation 0.07) and a specificity of 0.81 (range 0.73–0.90, standard deviation 0.05), with an area under the curve (AUC) score of 0.89 for the detection of LM/LMM.

In order to reduce the imbalance of the data, a further 28 additional benign tumors were provided by dermatologists, with the number of images per tumor varying between 4 and 103 images (608 images in total). These new patients were only considered for training purposes and they were not taken into account in Section 3.

In addition, these data did not provide any information regarding individual images. As these annotations were required in the next part of the work, each of these images was annotated by a specialist, with the help of a graphical interface designed for this purpose. Images with BCC tumors were not considered as the objective is to be able to discern LM/LMM. Whereas the patient labels were "benign" or "malignant", some of the images could not be classified into either of these two categories because they did not contain any of these pathology signs. For this particular reason, a "healthy" label was introduced to characterize them. Figure 2 provides an overview of these different data. As this study focused only on binary classification of malignant diseases, an annotation hierarchy was defined as follows:

- A "malignant" label: an image with at least some malignant tissues from LM/LMM tumors.
- A "benign" label: an image with no malignant tissues (either "benign" or "healthy" skin).

These annotated images amounted to 5277 images, divided equally between men and women. The "malignant" labels accounted for 44% of the annotated images, while the "benign" label accounted for 56%.

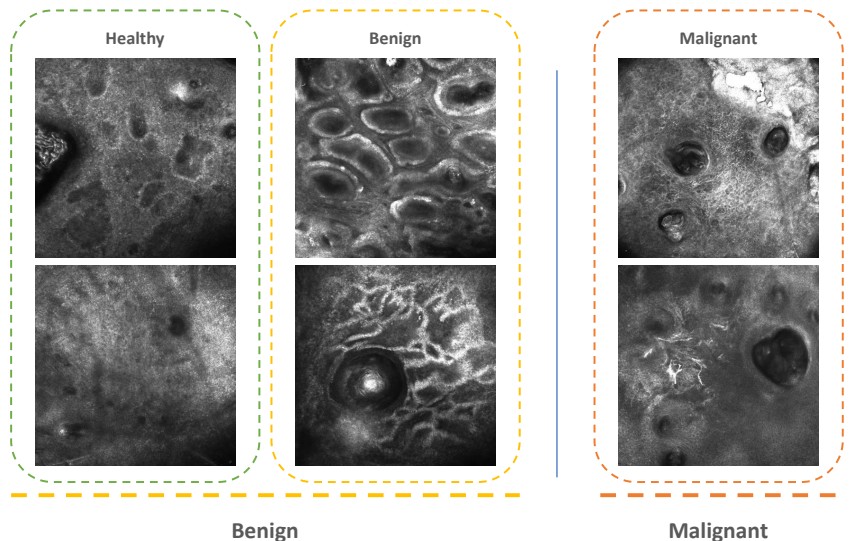

**Figure 2.** This figure shows several examples of images related to the three different types of tissue: namely "healthy", "benign" and "malignant". For this work purpose, the "healthy" and "benign" labels are merged under the "benign" label.

### 2.2. Feature Extraction Methods

In order to classify the data, a reduction of the image information had to be performed in a new feature space able to distinguish "malignant" from "benign" image types. According to the dermatologists, texture plays an important role in the differentiation of tissue types. The first part of this section focuses on handcrafted feature extractors based on texture from previous work [19]. The deep extraction methods applied to this context and inspired by a previous work on dermoscopy images [6] are then detailed. All of the feature extraction methods are listed in Table 1, and the next parts follow this table structure.

**Table 1.** The list of all of the feature extraction methods performed in this paper and their associated extracted number of features.

| Category | Name | Number of Features |
|---|---|---|
| Spatial | Haralick | 12 |
| | GLH + GLCM | 17 |
| Frequency | Fourier | 38 |
| | Wavelet | 39 |
| Transfer Learning | VGG-16 | 512 |
| | Inception-V3 | 2048 |
| | ResNet | 2048 |
| | Inception-ResNet | 1536 |

The "Spatial" extraction methods were based on spatial patterns of pixels, by use of GLH and GLCM. The method called "Haralick" refers to previous work based on texture features [20] and it uses the GLCM concept by computation of the twelve statistical characteristics listed in Table 2—GLCM

Features column. These characteristics were extracted along horizontal, vertical, and two diagonals. A mean was computed along these axes as the tissues are not oriented in space, and to reduce the number of features. A second method from previous work [19], called "GLH + GLCM" in this paper, expanded the first twelve initial characteristics of Haralick and it added five others based on GLH. In total, 17 features were extracted for each image, and all of the statistical properties extracted are listed in Table 2. The Haralick features extraction was performed using the "Mahotas" library [28] and histogram feature extraction was computed with help from the "Scipy" library [29].

**Table 2.** The statistical measures derived from GLCM and GLH, respectively, and extracted in order to perform "Spatial" extraction methods.

| GLCM Features | GLH Features |
| --- | --- |
| Angular Second Moment | Mean value |
| Difference Moment | Mean square deviation |
| Correlation | Skewness |
| Sum of Squares | Kurtosis |
| Inverse Difference Moment | Entropy |
| Summed Average | |
| Sum Variance | |
| Entropy | |
| Sum Entropy | |
| Difference Entropy | |
| Measure of Correlation 1 | |
| Measure of Correlation 2 | |

The second category of extraction methods, called "Frequency", refers to a set of methods based on frequency approaches. The first method of this category is called "Fourier" and is based on the Fourier transform. The main idea is to provide different levels of information as high frequency refers to high-contrast parts and low frequencies to homogeneous areas in the image. As the spectrum is symmetrical around the origin, only half of this spectrum was considered for the sake of computational efficiency. Then, a mean value was computed for all of the coefficients located at the same radial distance from the origin, at 22 different radius sizes between 0 and the diagonal size of the image. A previous paper [30] has also shown the relevance of this method in the context of textural images. Finally, 16 constant directions were taken from the origin of the power spectrum, and a mean value was computed for each of them [19]. The second method of this category is called "Wavelet" in this work and is based on Wavelet transform by use of a decomposition based on a Daubechies 4 that provides quite fine localization properties [19]. This decomposition was made at five scales, and only the four last scales were considered to compute coefficients. For each of them, three statistical measures were computed: the standard deviation, the energy, and the entropy.

The third category of methods investigated was in regard to deep learning methods and more specifically in regard to CNN, which are known to be well-suited methods for image classification, thanks to robust feature patterns [5]. Many architectures were used to address ImageNet challenges, and their associated performances were analyzed [31]. Instead of training this network from scratch, as we have data constraints and also computational constraints, we choose a Domain Adaptation approach for these models. Most of the papers to date have dealt with CNN trained on ImageNet [7], as this database contains thousands of classes and more than 14 million images, meaning that the extracted features from these networks can be used in various fields. As discussed in a previous work [32], Inception-V3 architecture pre-trained on ImageNet is thought to be the most relevant for medical applications. As RCM images can have various forms and can contain specifics details compared to other image modalities, this research compares the most well known CNN architectures: VGG-16 [33], Inception-V3 [34], ResNet [35] and Inception-ResNet [36], with accuracies of 0.71, 0.76, 0.78, and 0.80, respectively, on the ImageNet database [31]. This method involves the use of Transfer Learning, by removing the last layers devoted to the classification task, in order to obtain

a new representation of the image data as features. Furthermore, in order to reduce the number of features provided by the previous step, a global pooling layer on each activation layer was performed. For convenience, this whole method is called "Transfer Learning" in the next paragraphs and the names of the respective networks are used. The CNN computation was implemented using the "Keras" library [37].

### 2.3. Image-Level Decision

The image-level decision was the first level of classification achieved in this study, whereby the image classification must be carried out according to the two classes "malignant", as the positive class, and "benign". In order to satisfy this objective, the process consisted of using the image as a single instance that has several discriminant characteristics and sufficient information to allow its classification. As formulated by [38], such a problem can be set as a pair $X|y$, in which $X = \{x_1, x_2, \ldots, x_n\}$ is a vector characteristic for which $n$ is the number of features and $y$ the associated label. The task consisted of finding the existing relationship between $X$ and $y$, using a classification process.

To achieve this task, an extraction method (see Section 2.2) was applied to the images, depending on the currently evaluated method. The features were then normalized based on a standard score computation to make the classification task more accurate and robust [39]. This scaling was computed by subtracting the mean and then dividing by the standard deviation. The schematic outline in Figure 3 provides an overview of this process.

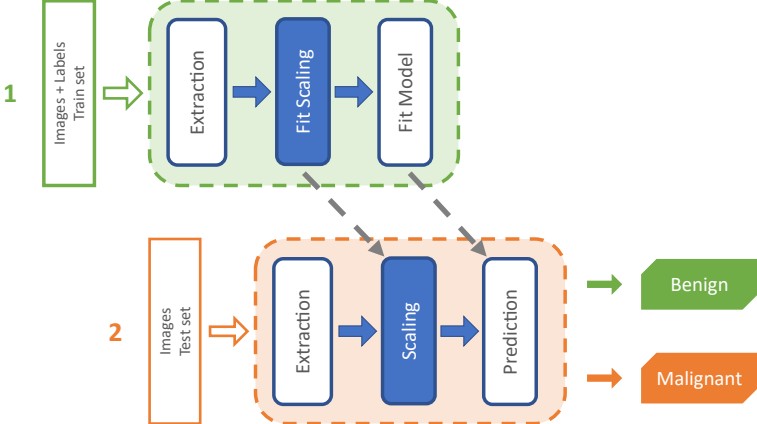

**Figure 3.** The classification process performed on the RCM images. The "Extraction" box refers to one of the Feature Extraction methods mentioned in Section 2.2. The "Fit Model" and "Prediction" boxes are related to the training and inference steps, respectively, of one of the trained and inferred models discussed in Section 2.3. The testing set was predicted based on two classes: "benign" and "malignant".

Finally, the classification was performed on scaled features using different models. In the first stage, CART was investigated as in a previous study in the same data context [19]. In a second stage, this study explored alternatives of simple trees based on ensemble methods (set of models instead of a single model) as the CART model tends to overfit. On the one hand, bagging methods were investigated by the use of random forest (RF) model [40] and extremely randomized trees (ERT) [41] as they both tend to remove the overfitting issues. Moreover, the ERT model are assumed to be more robust to noise than the RF model. On the other hand, the boosting method was considered by the use of the gradient boosting (GB) model [42] as the most common type of tree-based algorithm for most of the recent applications. Lastly, Support Vector Machine (SVM) models were evaluated that are known

to be suitable in multiples contexts [30,43]. As the relationship between the features and the expected outputs can be complex, SVM models were compared over linear and RBF kernels.

In addition, to provide the best performance with each of these models, a search in regard to their optimum hyperparameters was carried out (see Table 3).

**Table 3.** List of all of the classification models performed in this study and their referring evaluated hyper-parameters.

| Name | Parameter | Values |
|---|---|---|
| CART/RF/ERT | Maximum depth | [3, ∞] |
|  | Criterion | [Gini, Entropy] |
| GB | Maximum depth | [3, ∞] |
|  | Criterion | [Mean squared error, Mean absolute error] |
| SVM - Linear | C | [0.01, 0.1, 1, 10, 100, 1000] |
| SVM - RBF | C | [0.01, 0.1, 1, 10, 100, 1000] |
|  | Gamma | [0.01, 0.1, 1, 10, 100, 1000] |

### 2.4. Patient-Level Decision

This part relates to different ways to achieve classification at the patient-level based on the same two categories: "malignant", as the positive class, and "benign". With this assumption, a patient should be considered "malignant" if at least one image is considered to be "malignant". Additionally, this part needs to consider the varying number of samples per patient (as a reminder, the number of instances per patient can vary between 2 and 833 images).

In order to achieve this, the best combination of the feature extraction method and the classification model from Section 2.3 was used. The classification model provided two types of information for each image: the score was based on prediction probabilities and the decision (i.e., the class that achieved the best probability). In both cases, due to the varying number of images per patient, the information needs to be transformed into constant-size matrices to make a decision for existing and for new patients. At the score level, the structure was composed of patients, images, and scores of classes and transformed into a new matrix of size P × C, where P is the number of patients and C is the number of classes. A dynamic threshold was then used to adjust the positive class that maximizes the chosen metric. Multiple strategies can be used to achieve this:

- Mean—Allows the contribution of each instance on the patient to be retained
- Maximum—Retention of the best confidence prediction as to the trusted one.

At the decision level, the structure was composed of structure, combining patients, images, and scores of classes and transformed into a new matrix of size P*C, where P is the number of patients and C is the number of classes. Also, C refers to the probability vector of each decision between 0 and 1.

- At Least One—At least one positive decision to consider the input as positive (initial assumption)
- Dynamic—Find a dynamic threshold that minimizes false-positive decisions.

A global overview of the processing scheme of this method is presented in Figure 4. In a second stage, a number of multiple instance learning (MIL) concepts are implemented in this part, as they fit the issue at hand: a patient is constitutive of several instances (consider it as a bag) and a positive instance assumes that the patient should be positive. Furthermore, only the patient label is known, and the annotation step of individual images is time-consuming. Such a problem can be set as a pair $\{X|y\}$, in which $X = \{X^1, X^2, \ldots, X^b\}$ is a bag containing $b$ instances and each $X^b$ formulated as follows: $X^b = \{x_1^b, x_2^b, \ldots, x_n^b\}$, in which $n$ is the number of features and $y$ is the patient label [38]. Two ideas are developed regarding this context in the paragraphs that follow. In a first stage,

a Single-Instance Learning (SIL) classification is used in which a bag is considered as negative if all of the instances are considered to be negative, and positive if at least one of the instances has a positive label that fits our initial formulation of the patient label. In a second stage, the MI-SVM is an extension of Support Vector Machine (SVM) upon MIL theory and is employed in these due to the results of Section 2.3. These experiments are configured to function with a linear kernel due to the observation made by the experiments of Section 2.3. In this part, the experiments were implemented using the "MISVM" library [44].

In addition, to provide the best performance on each of these models, a search for their optimal hyper-parameters was carried out (see Table 4).

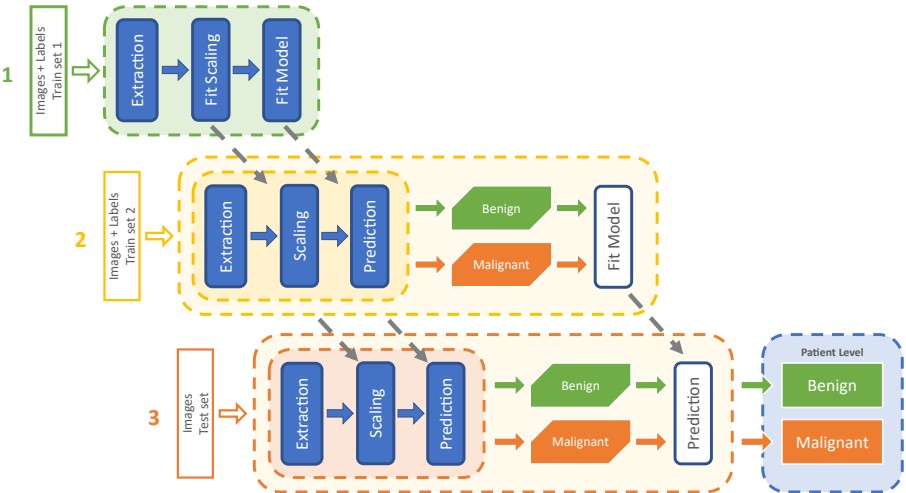

**Figure 4.** The classification process performed on RCM patients in which the first part of the process remained the same as in Section 2.3. The "Fit Model" and "Prediction" boxes refer to the decision and score level methods discussed in Section 2.4. The testing set was predicted for the patients based on "benign" and "malignant" classes.

**Table 4.** List of all of the classification models and referring evaluated hyper-parameters.

| Category | Name | Parameter | Values |
|---|---|---|---|
| Score | Mean | - | [] |
| | Maximum | | |
| Decision | At Least One | - | [] |
| | Dynamic | | |
| MIL | SIL | C | [0.01, 0.1, 1, 10, 100, 1000] |
| | MI-SVM | | |

## 3. Results

### 3.1. Validation and Evaluation Metric

The validation protocol remains the same for each of these experiments, based on a nested cross-validation that is known to be less biased than a simple cross-validation scheme [45]. This protocol allows (1) cross-validation of hyper-parameters and (2) objective evaluation of the prediction models. Each of the cross-validation step is based on a K-fold strategy with a $k$ value of 4 on the testing loop and 2 on the validation loop. Also, each time, the patients are separated and balanced as best as possible based on the image labels. In order to achieve an objective evaluation, each data cluster remains the same for the experiments in a given section (refer to Sections 2.3

and 2.4). Moreover, each experiment is validated and evaluated using a $F_1$ score metric, as it is statistically suitable for unbalanced populations in comparison with accuracy, and it represents in a single value both recall and precision information. In addition, standard deviation is computed to analyse the stability of models along the nested cross-validation. For this purpose, we used the "Scikit Learn" library for Machine Learning classification, validation, and metric [46].

### 3.2. Experiments and Discussion

The results of Section 2.3 regarding image-level decisions were performed only on the labeled images and are listed in Table 5. The most suitable handcrafted extraction method was based on the "Wavelet" method combined with the "SVM-Linear" model, reaching an weighted $F_1$ score of 0.74 with stable performance of 0.04. In general, all of the handcrafted methods performed in a quite similar and stable way, varying from 0.69 to 0.74 for the $F_1$ score and 0.03 to 0.07 for the deviation. In addition, the Transfer Learning-based feature extraction reached higher scores with the "SVM-Linear" model, in particular with the "Inception-ResNet" architecture, which achieved a weighted $F_1$ score of 0.82 with a deviation of 0.04. With this same model, the Transfer Learning feature extraction methods varied from 0.76 to 0.82, with a deviation range from 0.02 to 0.04. By contrast, all of these architectures were poorly processed by the "SVM-RBF" model and can be explained by the high-dimensionality of the extracted features that resulted in overfitting despite the cross-validation of the regularization term. In this situation, the "VGG-16" architecture was more suitable with only 512 features than the remaining architectures providing 1536 or 2,048 features. In regards to tree-based models, the CART model weighted $F_1$ score varies between 0.69 and 0.71 (deviation between 0.03 and 0.05) and 0.58 to 0.64 (deviation between 0.02 and 0.12), respectively for handcrafted methods and transfer learning methods. On the other hand, the gradient boosting (GB) model weighted $F_1$ score varies from 0.67 to 0.73 (deviation between 0.04 and 0.07) and 0.78 to 0.81 (deviation between 0.04 and 0.05). The above two sentences can be explained by an overfit in a high dimensional situation for the CART model and in a low dimensional situation for the GB model. In opposition, the random forest (RF) and extremely randomized trees (ERT) models were homogeneous along with handcrafted and transfer learning features as they are less prone to overfitting in both low and high dimensional feature spaces. To sum up the aforementioned results, the rest of this article retains only the best combination, with "Inception-ResNet" as the feature extraction methods and the Linear SVM as the classification model.

**Table 5.** List of the results based on combinations of features extraction methods from Section 2.2 and the classification models from Section 2.3 evaluated over a weighted $F_1$ score based on benign and malignant classifications.

| | Classifier Type | | | | | |
| --- | --- | --- | --- | --- | --- | --- |
| | **CART** | **RF** | **ERT** | **GB** | **SVM - Linear** | **SVM - RBF** |
| **Haralick** | $0.71 \pm 0.05$ | $0.71 \pm 0.04$ | $0.71 \pm 0.05$ | $0.67 \pm 0.05$ | $0.71 \pm 0.05$ | $0.70 \pm 0.07$ |
| **GLH + GLCM** | $\mathbf{0.71 \pm 0.03}$ | $0.71 \pm 0.03$ | $0.72 \pm 0.05$ | $0.67 \pm 0.07$ | $0.70 \pm 0.05$ | $0.72 \pm 0.04$ |
| **Fourier** | $0.69 \pm 0.05$ | $0.70 \pm 0.03$ | $0.70 \pm 0.04$ | $0.69 \pm 0.04$ | $0.73 \pm 0.03$ | $0.69 \pm 0.06$ |
| **Wavelet** | $0.70 \pm 0.03$ | $0.72 \pm 0.04$ | $0.74 \pm 0.05$ | $0.73 \pm 0.05$ | $0.74 \pm 0.04$ | $\mathbf{0.72 \pm 0.03}$ |
| **VGG-16** | $0.58 \pm 0.12$ | $0.73 \pm 0.03$ | $0.76 \pm 0.06$ | $0.78 \pm 0.04$ | $0.76 \pm 0.03$ | $0.64 \pm 0.20$ |
| **Inception-V3** | $0.63 \pm 0.04$ | $0.74 \pm 0.06$ | $0.78 \pm 0.04$ | $0.79 \pm 0.05$ | $0.79 \pm 0.03$ | $0.44 \pm 0.04$ |
| **ResNet** | $0.62 \pm 0.08$ | $0.75 \pm 0.05$ | $0.78 \pm 0.02$ | $\mathbf{0.81 \pm 0.05}$ | $0.79 \pm 0.02$ | $0.43 \pm 0.05$ |
| **Inception-ResNet** | $0.64 \pm 0.02$ | $\mathbf{0.76 \pm 0.03}$ | $\mathbf{0.79 \pm 0.05}$ | $\mathbf{0.81 \pm 0.05}$ | $\mathbf{0.82 \pm 0.04}$ | $0.44 \pm 0.06$ |

This paragraph focuses on the methods implemented to reach the patient diagnosis (see Section 2.4) and it relates to the initial RCM data (including unlabeled images) used previously to evaluate specialists [27]. All these experimental results are listed in Table 6 and discussed below.

Firstly, the methods performance varied between 0.61 and 0.84 in terms of the $F_1$ score for Malignancy. The "At Least One" method achieved poor performance due to prediction errors over the "benign" class. This problem can be solved by the use of a dynamic activation threshold for decisions to minimize the risk of false-positives, at the cost of resulting in an ethical consideration of this method in the clinical context. Secondly, the methods based on the score are almost the same and varied from 0.76 to 0.83 for the $F_1$ score for Malignancy. By contrast, with these results, the standard deviations remained reasonable, varying from 0.03 to 0.06. Finally, MIL was also evaluated, and a substantial difference was noted between the SIL and the MI-SVM. Indeed, the SIL assumption yielded similar results with the decision based on the "At Least One" method, due to an insufficient ability to discriminate on the same "benign" class. By contrast, the MI-SVM yielded a number of good results, with an $F_1$ score of 0.82. Both methods are stable, with a deviation that only varied from 0.02 to 0.04. Poor results with the "At Least One" and the SIL methods can be due to a lack of discriminative information provided by the "Inception-ResNet" for these methods.

**Table 6.** Results for the patient-level classification for Malignancy (LM/LMM and BCC) according to the different methods from Section 2.4. For Malignancy and LM/LMM, the table provides a weighted average $F_1$ score and individual $F_1$ score for the benign and the malignant classes.

| Category | Name | Malignancy—$F_1$ Score | | |
|---|---|---|---|---|
| | | **Weighted** | **Benign** | **Malignant** |
| Decision | At Least One | $0.61 \pm 0.06$ | $0.32 \pm 0.07$ | $0.79 \pm 0.05$ |
| | Dynamic | $\mathbf{0.84 \pm 0.03}$ | $0.78 \pm 0.07$ | $\mathbf{0.87 \pm 0.02}$ |
| Score | Mean | $0.83 \pm 0.03$ | $0.78 \pm 0.08$ | $0.87 \pm 0.02$ |
| | Maximum | $0.76 \pm 0.04$ | $0.68 \pm 0.03$ | $0.80 \pm 0.05$ |
| MIL | SIL | $0.70 \pm 0.04$ | $0.50 \pm 0.10$ | $0.83 \pm 0.03$ |
| | MI-SVM | $0.82 \pm 0.02$ | $\mathbf{0.78 \pm 0.05}$ | $0.84 \pm 0.02$ |

Following previous results, this paragraph discusses in detail the results of supervised "Dynamic" decision threshold and MIL based on "MI-SVM" methods over the Malignancy (meaning BCC and LM/LMM) and LM/LMM as the cited clinical study does [27]. Table 7 provides $F_1$ score, precision, and recall based on these experiments. As the classification is binary, recall of the positive class refers to the sensitivity and recall of the negative class refers to the Specificity. The "Dynamic" method achieves scores of $0.89 \pm 0.03$ sensitivity and $0.75 \pm 0.07$ specificity for Malignancy; $0.88 \pm 0.04$ sensitivity and $0.75 \pm 0.07$ specificity for LM/LMM pathologies. The "MI-SVM" method achieves scores of $0.80 \pm 0.02$ sensitivity and $0.84 \pm 0.05$ specificity for Malignancy; $0.78 \pm 0.07$ sensitivity and $0.84 \pm 0.07$ specificity for LM/LMM pathologies. The "Dynamic" method provides more emphasis on sensitivity while "MI-SVM" provides a good specificity. These methods are quite comparable to the evaluation of the dermatologists, reaching 0.80 of sensitivity and 0.81 of specificity, but less homogeneous compared to them.

Finally, Figure 5 provides receiver operating characteristic (ROC) curves for both malignancy and LM/LMM pathologies on "Dynamic" and "MI-SVM" methods. In the context of Malignancy evaluation, the measured AUC is 0.89 for "MI-SVM" and 0.88 for "Dynamic". For LM/LMM evaluation, the measured AUC is 0.88 for "MI-SVM" and 0.87 for "Dynamic". In the same context of LM/LMM lesions, the experts obtained an AUC score of 0.89, so close to the previous two methods. Apart from this, Figure 6 provides some misleading images: the RCM images in the center belongs to the same patient (image c and d) with similar patterns and homogeneous information while the RCM images on the outside parts of the figure contain hair, artifacts, tricky patterns or nonhomogeneous information (image a, b, e, and f). Also, the images on the bottom left and the bottom right (image b and e) of the figure are examples of images were experts will use stacks of images to make their decision and where the currently developed methods only use a single image.

**Table 7.** Detailed results for the patient-level classification for the Decision method based on a Dynamic threshold and MIL based on the MI-SVM assumption. The table provides the $F_1$ score, Precision, and Recall for the benign and malignant classes with these methods.

| Name | Label | Malignancy | | | LM/LMM | | |
|---|---|---|---|---|---|---|---|
| | | $F_1$ Score | Precision | Recall | $F_1$ Score | Precision | Recall |
| Dynamic | Benign | $0.78 \pm 0.07$ | $0.81 \pm 0.08$ | $0.75 \pm 0.07$ | $0.79 \pm 0.06$ | $0.82 \pm 0.07$ | $0.75 \pm 0.07$ |
| | Malignant | $0.87 \pm 0.02$ | $0.85 \pm 0.03$ | $0.89 \pm 0.03$ | $0.86 \pm 0.03$ | $0.83 \pm 0.03$ | $0.88 \pm 0.04$ |
| | Weighted | $0.84 \pm 0.03$ | $0.84 \pm 0.03$ | $0.84 \pm 0.03$ | $0.83 \pm 0.03$ | $0.83 \pm 0.03$ | $0.83 \pm 0.03$ |
| MI-SVM | Benign | $0.78 \pm 0.02$ | $0.72 \pm 0.08$ | $0.84 \pm 0.07$ | $0.78 \pm 0.05$ | $0.73 \pm 0.07$ | $0.84 \pm 0.07$ |
| | Malignant | $0.84 \pm 0.02$ | $0.89 \pm 0.05$ | $0.80 \pm 0.05$ | $0.82 \pm 0.03$ | $0.87 \pm 0.06$ | $0.78 \pm 0.07$ |
| | Weighted | $0.82 \pm 0.02$ | $0.82 \pm 0.02$ | $0.83 \pm 0.02$ | $0.80 \pm 0.03$ | $0.80 \pm 0.03$ | $0.81 \pm 0.02$ |

**Malignancy ROC curves**

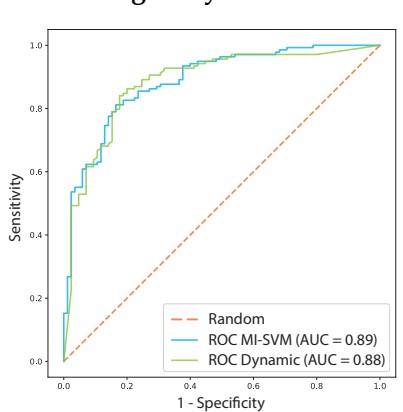

**LM/LMM ROC curves**

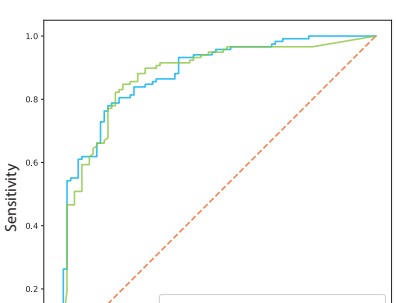

**Figure 5.** On the **left**, the ROC curves for Malignancy with the Dynamic and MI-SVM methods. On the **right**, the ROC curves for LM/LMM with the Dynamic and MI-SVM methods.

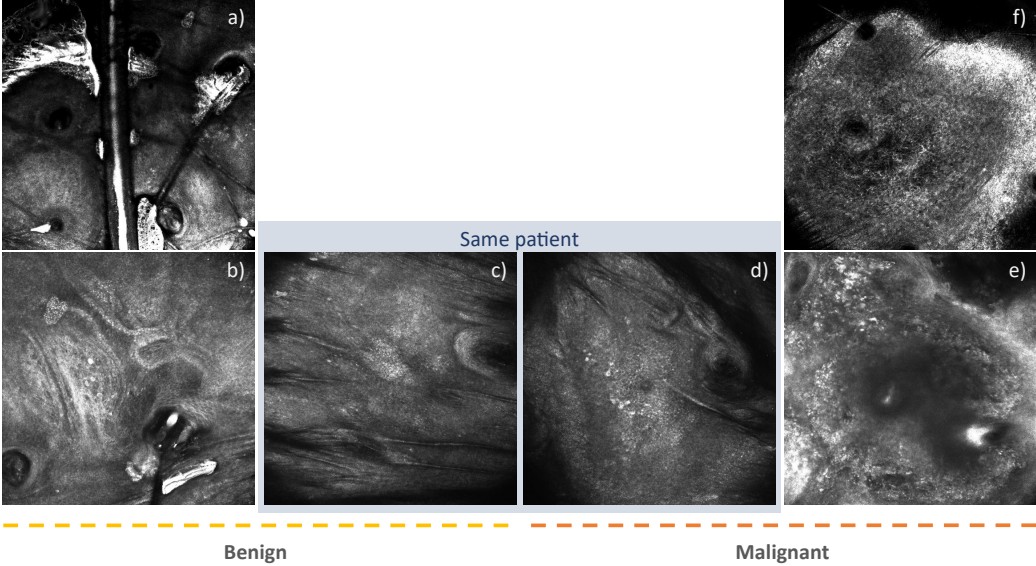

**Figure 6.** Examples of RCM images that mislead the classifier with the highest $F_1$ score (Inception-ResNet + SVM − Linear). On the left part (**a–c**), RCM images belonging to the benign label and classified as malignant. On the right part (**d–f**), RCM images belonging to the malignant label and classified as benign. In the center of the figure (**c,d**), two RCM images of the same patient belonging to benign and malignant labels.

## 4. Conclusions

This research investigated the classification of malignant tumors and particularly LM/LMM pathologies at the image-level and at the patient-level. Firstly, at the image-level, an analysis was performed of previous research and proposed methods based on Transfer Learning showing that the Inception-ResNet architecture trained on the ImageNet database is quite relevant for classifying RCM images, with a weighted $F_1$ score of 0.82 between benign and malignant labels. Secondly at the patient-level, supervised decision methods were compared to MIL methods based on feature extraction with the Inception-ResNet architecture. On the first hand, the classification of malignancy over patients achieves a weighted $F_1$ score of 0.84 with an AUC score of 0.88 using the supervised method based on a dynamic threshold. It achieves a weighted $F_1$ score of 0.82 with an AUC score of 0.89 using the MIL method based on MI-SVM. On the other hand, the classification of LM/LMM over patient achieves a weighted $F_1$ score of 0.83 (Sensitivity 0.88/Specificity 0.75) with an AUC score of 0.87 using the supervised method based on a dynamic threshold. It achieves a weighted $F_1$ score of 0.80 (Sensitivity 0.78/Specificity 0.84) with an AUC score of 0.88 using the MIL method based on MI-SVM. Both techniques are relevant compared to the evaluation of dermatologists, reaching 0.80 of sensitivity and 0.81 of specificity with an AUC score of 0.89. Furthermore, the supervised method based on the dynamic threshold has more sensitivity that can be relevant in the medical context.

To conclude, the RCM is an operator dependent technique that can be improved by use of computer-aided diagnosis methods that can be particularly useful to confirm the diagnosis and to help the interpretation of the images for clinicians that are not expert. Further development of these findings should focus on the enhancement of the image classification through feature extraction enhancement, in particular by investigation of fine-tuning of the CNN architecture. In another step, the patient decision should be improved by working on the score meaning through score calibration methods and by rethinking the way the decision is taken as a result of these.

**Author Contributions:** Investigation, R.C.; Methodology, R.C.; Project administration, F.M.; Resources, J.-L.P. and E.C.; Supervision, A.M. and F.M.; Writing—original draft, R.C.; Writing—review and editing, R.C., A.M., J.-L.P., E.C. and F.M. All authors have read and agreed to the published version of the manuscript.

**Funding:** This research was funded by the Conseil Regional de Bourgogne Franche-Comte of France and the European Regional Development Fund (ERDF).

**Conflicts of Interest:** The authors declare no conflict of interest.

**Acknowledgments:** We thank Perrot and Cinotti for their work on the data and the permission granted to exploit it.

## Abbreviations

The following abbreviations are used in this manuscript:

| | |
|---|---|
| AUC | Area under the curve |
| BCC | Basal cell carcinoma |
| CART | Classification and regression trees |
| CNN | Convolutional neural network |
| DEJ | Dermal-epidermal junction |
| ERT | Extremely randomized trees |
| GB | Gradient boosting |
| GGD | Generalized Gaussian distribution |
| GLH | Gray level histogram |
| GLCM | Gray level co-occurrence matrix |
| LM/LMM | Lentigo maligna/lentigo maligna melanoma |
| MIL | Multiple instance learning |
| RCM | Reflectance confocal microscopy |
| RF | Random forest |
| ROC | Receiver operating characteristic |
| SIL | Single-Instance Learning |
| SVM | Support Vector Machine |

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
