# Peer review of "Classification of Lentigo Maligna at Patient-Level by Means of Reflectance Confocal Microscopy Data"

_applsci, doi:10.3390/app10082830_

Round 1

Reviewer 1 Report

Thank you for updating your manuscript. I have nothing more to say.

Author Response

Dear reviewer,

Thank you for taking the time to report the manuscript “Classification of lentigo maligna at patient-level by means of reflectance confocal microscopy data” and thank you for all your constructive remarks.

Kindly regards,

Romain CENDRE

Reviewer 2 Report

This is a nice piece of work and can be published after minor revisions.

  • Please improve the end of the abstract. Lot of numbers are listed here, but the reader would be more interested about the potential applicability of the results and its comparison with existing methods. 
  • The stups in fig. 1 could be merged to one setup, showing all of the optical paths in one single drawing.

Author Response

Dear reviewer,

Thank you for taking time to report the manuscript “Classification of lentigo maligna at patient-level by means of reflectance confocal microscopy data”. In the following paragraphs, we answer point by point to your remarks.

  1. Reviewer: This is a nice piece of work and can be published after minor revisions.
    Answer: Thank you for this positive remark and we will try to answer as best as we can!
  1. Reviewer: Please improve the end of the abstract. Lot of numbers are listed here, but the reader would be more interested about the potential applicability of the results and its comparison with existing methods.
    Answer:   We agree on these changes and write it at the following: “Abstract: Reflectance confocal microscopy is an appropriate tool for the diagnosis of lentigo maligna. Compared with dermoscopy, this device can provide abundant information as a mosaic and/or a stack of images. In this particular context, the number of images per patient varied between 2 and833 images and the objective, ultimately, is to be able to discern between benign and malignant classes.  First, this paper evaluated classification at the image level, with the help of handcrafted methods derived from the literature and transfer learning methods. The transfer learning feature extraction methods outperformed the handcrafted feature extraction methods from literature, with aF1score value of 0.82. Secondly, this work proposed patient-level supervised methods based on image decisions and a comparison of these with multi-instance learning methods.  This study achieved comparable results to those of the dermatologists, with an AUC score of 0.87 for supervised patient diagnosis and an AUC score of 0.88 for multi-instance learning patient diagnosis. According to these results, computer-aided diagnosis methods presented in this paper could be easily used in a clinical context to save time or confirm a diagnosis and can be oriented to detect images of interest. Also, this methodology can be used to serve future works based on multimodality.
  2. Reviewer: The stups in fig. 1 could be merged to one setup, showing all of the optical paths in one single drawing.
    Answer: We agree with this suggestion. The figure has been updated to fit your remark, and the caption is now “Figure 1. Principle of the RCM designed by Marvin Minsky [9]. The light source is transmitted through a pinhole and focused on the sample through an objective. Then the in-focus reflected light is collected by the detector with the help of the second pinhole.”. Also, we updated the main text: “However, dermoscopy imaging devices only provide surface and chromatic information.  To overcome this limitation, reflectance confocal microscopy (RCM) modality is another type of imaging technique used by dermatologists that provides high-resolution images of the skin on a micrometer scale. Furthermore, this modality can provide structural information at different depths of the skin by adjustment of the wavelength properties and the focal point [8]. The RCM device was first designed by Marvin Minsky [9]. The principle of this device is to emit and focus a low power laser on a specific point of the skin, then the light from this spot is reflected and collected through an objective and a pinhole that allows only the light from the in-focus plane to reach the detector (see Figure 1). In this situation, the illuminated point and the detector aperture have confocal (a contraction of conjugate focal) planes [10].  The main interest of this focal point/plane is to provide only tissue information from a specifically chosen depth.  Different factors can affect the depth:  illumination wavelength, illumination power, reflective and scattering properties of the skin.

We also added some additional text about the dataset on the Data section for clarification:

These data include 223 lesions from 201 patients, for a total of 7,846 RCM images. Generally, one lesion is equal to one patient so we will discuss of a lesion as a patient in the next paragraphs. Each of these cases varies between 2 and 833 images (with a mean of 35 and a standard deviation of 64).”

And modification on conclusion following the ones on abstract:

To conclude, the RCM is an operator dependent technique that can be improved by use of computer-aided diagnosis methods that can be particularly useful to confirm the diagnosis and to help the interpretation of the images for clinicians that are not expert. Further development of these findings should focus on the enhancement of the image classification through feature extraction enhancement, in particular by investigation of fine-tuning of the CNN architecture.  In another step, the patient decision should be improved by working on the score meaning through score calibration methods and by rethinking the way the decision is taken as a result of these.

In addition, you will find the reviewed file with highlighted changes attached to these answers.

We hope that the corrections made to the document meet your expectations,

Romain CENDRE

This manuscript is a resubmission of an earlier submission. The following is a list of the peer review reports and author responses from that submission.

Round 1

Reviewer 1 Report

Well tested and considered.
Decision tree-based methods like the random forest, xgboost could be included in the list of candidate classifiers as they are already in the scikit-learn library.
Showing some of the misclassified images could be helpful for readers to understand the difficulties of the research goals.

Strongly suggest using grammar checkers like Grammarly (free version available) to avoid at least simple mistakes like repeating "convolutional neural network" twice in the third paragraph on page 5.

Author Response

Dear reviewer,

Thank you for taking time to report the manuscript “Classification of lentigo maligna at patient-level by means of reflectance confocal microscopy data”. In the following paragraphs, we answer point by point to your remarks.

Reviewer: Well tested and considered.
Answer: Thank you for this positive remark, we are glad that you considered our work! Reviewer: Decision tree-based methods like the random forest, xgboost could be included in the list of candidate classifiers as they are already in the scikit-learn library.
Answer: We agree with your suggestions. Initially, random forest was first investigated in this study, with extra trees to deal with a high number of features. As their principles remain close, and the results of extra trees were as expected better on classification over transfer learning features, we only mentioned it to avoid such many models. We added the results related to this model. Gradient boosting was not considered, and we add results to the paper as an alternative ensemble model to consider. The results are computed on the same folds as the other models. The following text have been added in the methods section: "Finally, the classification was performed on scaled features using different models. In the first stage, CART was investigated as in a previous study in the same data context [18]. In a second stage, this study explored alternatives of simple trees based on ensemble methods (set of models instead of a single model) as the CART model tends to overfit. On the one hand, bagging methods were investigated by the use of random forest (RF) model [39] and extremely randomized trees (ERT) [40] as they both tend to remove the overfitting issues. Moreover, the ERT model are assumed to be more robust to noise than the RF model. On the other hand, the boosting method was considered by the use of the gradient boosting (GB) model [41] as the most common type of tree-based algorithm for most of the recent applications. Lastly, Support Vector Machine (SVM) models were evaluated that are known to be suitable in multiples contexts [29,42]. As the relationship between the features and the expected outputs can be complex, SVM models were compared over linear and RBF kernels." and in the results: "In regards to tree-based models, the CART model weighted F1 score varies between 0.69 and 0.71 (deviation between 0.03 and 0.05) and 0.58 to 0.64 (deviation between 0.02 and 0.12), respectively for handcrafted methods and transfer learning methods. On the other hand, the GB model weighted F1 score varies from 0.67 to 0.73 (deviation between 0.04 and 0.07) and 0.78 to 0.81 (deviation between 0.04 and 0.05). The above two sentences can be explained by an overfit in a high dimensional situation for the CART model and in a low dimensional situation for the GB model. In opposition, the RF and ERT models were homogeneous along with handcrafted and transfer learning features as they are less prone to overfitting in both low and high dimensional feature spaces. To sum up the aforementioned results, the rest of this article retains only the best combination, with “Inception-ResNet” as the feature extraction methods and the Linear SVM as the classification model." Reviewer: Showing some of the misclassified images could be helpful for readers to understand the difficulties of the research goals.
Answer: We agree with this suggestion. Some of the misleading images have been added to the paper in Experiments and Discussion. The last paragraph of the Results section is also completed with explanations: "Finally, Figure 5 provides receiver operating characteristic (ROC) curves for both malignancy and LM/LMM pathologies on “Dynamic” and “MI-SVM” methods. In the context of Malignancy evaluation, the measured AUC is 0.89 for “MI-SVM” and 0.88 for “Dynamic”. For LM/LMM evaluation, the measured AUC is 0.88 for “MI-SVM” and 0.87 for “Dynamic”. In the same context of LM/LMM lesions, the experts obtained an AUC score of 0.89, so close to the previous two methods. Apart from this, Figure 6 provides some misleading images: the RCM images in the center belongs to the same patient (image c and d) with similar patterns and homogeneous information while the RCM images on the outside parts of the figure contain hair, artifacts, tricky patterns or nonhomogeneous information (image a, b, e, and f). Also, the images on the bottom left and the bottom right (image b and e) of the figure are examples of images were experts will use stacks of images to make their decision and where the currently developed methods only use a single image." Reviewer: Strongly suggest using grammar checkers like Grammarly (free version available) to avoid at least simple mistakes like repeating "convolutional neural network" twice in the third paragraph on page 5.
Answer: Thank you for noticing that, we have made the change in the text for the "convolutional neural network" repeating term and corrected some other mistakes.

In addition, you will find a difference file attached.

We hope we sufficiently answered to your remarks,

Romain CENDRE

Reviewer 2 Report

1. Currently, gold standard for diagnosis is not (only) histology, but much more clinic plus CPC (clinicopathological correlation).

2. With regard to Point 1 sensitivity levels in this manuscript of around 0.88 and specificity of around 0.75 are from a practical approach not helpful and working.

3. One of/or "the" main clinical problem(s) is the very frequent collision of events, namely different diagnoses in such constellation as melanoma in situ, type Lentigo maligna. You will have verrucae seborrhoicae, type Lentigo senilis, in situ squamous cell carcinoma, type (pigmented) aktinic keratosis, or remnants of some melanocytic nevi, mostly type Miescher or less common Zitelli all together in variable frequency in such instance.

4. "Lentigo maligna" is classical euphemism, exact diagnosis is melanoma in situ, type lentigo maligna, or invasive melanoma with Clark Level xy and Breslow index yx in mm, Type Lentigo maligna melanoma, respectively.

Author Response

Dear reviewer,

Thank you for taking time to report the manuscript “Classification of lentigo maligna at patient-level by means of reflectance confocal microscopy data”. In the following paragraphs, we answer point by point to your remarks.

Reviewer: Currently, gold standard for diagnosis is not (only) histology, but much more clinic plus CPC (clinicopathological correlation).
Answer: We agree with the Reviewer. Our standard for the diagnosis is clinicopathological correlation and the diagnosis of the skin lesions evaluated in this study was based on clinicopathological correlation. This aspect has been added in the manuscript. Reviewer: With regard to Point 1 sensitivity levels in this manuscript of around 0.88 and specificity of around 0.75 are from a practical approach not helpful and working.
Answer: Our results are not so far from those obtained by experts on the same series of lesions. The mean sensitivity for LM/LMM based on RCM images evaluated by the experts was 0.80 (range 66-90, SD 7), and the mean specificity was 0.81(range 73-90, SD 5) (Dermoscopy vs. reflectance confocal microscopy for the diagnosis of lentigo maligna, JEADV 2018 Aug;32(8):1284-1291.). These data have been added. Reviewer: One of/or "the" main clinical problem(s) is the very frequent collision of events, namely different diagnoses in such constellation as melanoma in situ, type Lentigo maligna. You will have verrucae seborrhoicae, type Lentigo senilis, in situ squamous cell carcinoma, type (pigmented) aktinic keratosis, or remnants of some melanocytic nevi, mostly type Miescher or less common Zitelli all together in variable frequency in such instance.
Answer: Collisions were excluded by our series and this aspect has been added in the manuscript. Reviewer: "Lentigo maligna" is classical euphemism, exact diagnosis is melanoma in situ, type lentigo maligna, or invasive melanoma with Clark Level xy and Breslow index yx in mm, Type Lentigo maligna melanoma, respectively.
Answer: You are completely right, “lentigo maligna” is an improper term. However, we decided to keep this terminology because this term is used in our hospitals in clinical routine and is still used in the last WHO clinic-pathological classification of skin tumors (WHO Classification of Skin Tumours. Fourth Edition Elder DE, Massi D, Scolyer R, Willemze R). Moreover, this term was used in the main RCM studies about this tumor (see for example: The impact of in vivo reflectance confocal microscopy on the diagnostic accuracy of lentigo maligna and equivocal pigmented and nonpigmented macules of the face.J Invest Dermatol. 2010) and is still used in the last clinical studies concerning this topic. See for example:

1: Navarrete-Dechent C, Cordova M, Aleissa S, Liopyris K, Dusza SW, Kose K, Busam KJ, Hollman T, Lezcano C, Pulitzer M, Chen CJ, Lee EH, Rossi AM, Nehal KS. Lentigo maligna melanoma mapping using reflectance confocal microscopy correlates with staged excision: A prospective study. J Am Acad Dermatol. 2019 Dec 5. pii: S0190-9622(19)33150-0. doi: 10.1016/j.jaad.2019.11.058. [Epub ahead of print] PubMed PMID: 31812621.

2: Hao T, Meng XF, Li CX. A meta-analysis comparing confocal microscopy and dermoscopy in diagnostic accuracy of lentigo maligna. Skin Res Technol. 2019 Dec doi: 10.1111/srt.12821. [Epub ahead of print] PubMed PMID: 31811677.

3: Robinson M, Primiero C, Guitera P, Hong A, Scolyer RA, Stretch JR, Strutton G, Thompson JF, Soyer HP. Evidence-Based Clinical Practice Guidelines for the Management of Patients with Lentigo Maligna. Dermatology. 2019 Oct 22:1-6. doi: 10.1159/000502470. [Epub ahead of print] Review. PubMed PMID: 31639788.

4: Farnetani F, Manfredini M, Chester J, Ciardo S, Gonzalez S, Pellacani G. Reflectance confocal microscopy in the diagnosis of pigmented macules of the face: differential diagnosis and margin definition. Photochem Photobiol Sci. 2019 May 15;18(5):963-969. doi: 10.1039/c8pp00525g. Review. PubMed PMID: 30938378.  

5: Navarrete-Dechent C, Liopyris K, Cordova M, Busam KJ, Marghoob AA, Chen CJ. Reflectance Confocal Microscopic and En Face Histopathologic Correlation of the Dermoscopic "Circle Within a Circle" in Lentigo Maligna. JAMA Dermatol. 2018 Sep 1;154(9):1092-1094. doi: 10.1001/jamadermatol.2018.2216. PubMed PMID: 30046812; PubMed Central PMCID: PMC6631332.

6: Mataca E, Migaldi M, Cesinaro AM. Impact of Dermoscopy and Reflectance Confocal Microscopy on the Histopathologic Diagnosis of Lentigo Maligna/Lentigo Maligna Melanoma. Am J Dermatopathol. 2018 Dec;40(12):884-889. doi: 10.1097/DAD.0000000000001212. PubMed PMID: 29933314.

Penultimate paragraph in the introduction was changed: "The scope of this work is to detect malignant tumors and particularly lentigo maligna/lentigo maligna melanoma (LM/LMM) (the most common type of facial melanoma) in RCM images and to help specialists reach a diagnosis based on these images at the patient-level. The previous feature extraction methods and extraction through different convolutional neural network (CNN) architectures were investigated first. As wavelet decomposition and reduction through GGD [20] were shown to be irrelevant in our data context [21], this work did not focus on any of these methods. A comparison of several classification models on full-size images was then carried out to estimate the relevance of these methods. Also, this study will employ the term “lentigo maligna” instead of the term “melanoma in situ type lentigo maligna” as it was used by the last RCM clinical studies [22–25] and for the sake of simplicity."

In addition, you will find as an attached file the differences between original submission and new one.

We hope we sufficiently answered to your remarks,

Elisa CINOTTI

Round 2

Reviewer 2 Report

Nothing new to add with regard to last review.